# Understanding the Experiences of Clinicians Accessing Electronic Databases to Search for Evidence on Pain Management Using a Mixed Methods Approach

**DOI:** 10.3390/healthcare11121728

**Published:** 2023-06-12

**Authors:** Vanitha Arumugam, Joy C. MacDermid, Dave Walton, Ruby Grewal

**Affiliations:** 1Chronic Pain Management Program, St. Joseph’s Healthcare London, London, ON N6A 4V2, Canada; 2Department of Surgery, University of Western Ontario, London, ON N6A 3K7, Canada; 3Hand and Upper Limb Centre Clinical Research Laboratory, St. Joseph’s Health Centre, London, ON N6A 4V2, Canada; 4School of Physical Therapy, University of Western Ontario, London, ON N6A 3K7, Canada

**Keywords:** mixed methods, literature review, CSR, qualitative interviews

## Abstract

The act of searching and retrieving evidence falls under the second step of the EBP process—tracking down the best evidence. The purpose of this study is to understand the competencies of clinicians accessing electronic databases to search for evidence on pain management using a mixed methods approach. Thirty-seven healthcare professionals (14 occupational therapists, 13 physical therapists, 8 nurses, and 2 psychologists) who are actively involved in pain management were included. This study involved two parts (a qualitative and a quantitative part) that ran in parallel. Participants were interviewed using a semi-structured interview guide (qualitative data); data were transcribed verbatim. During the interview, participants were evaluated in comparison to a set of pre-determined practice competencies using a chart-stimulated recall (CSR) technique (quantitative data). CSR was scored on a 7-point Likert scale. Coding was completed by two raters; themes across each of the competencies were integrated by three raters. Seven themes evolved out of the qualitative responses to these competencies: formulating a research question, sources of evidence accessed, search strategy, refining the yield, barriers and facilitators, clinical decision making, and knowledge and awareness about appraising the quality of evidence. The qualitative results informed an understanding of the strengths and weaknesses in the competencies evaluated. In conclusion, using a mixed methods approach, we found that clinicians were performing well with their basic literature review skills, but when it came to advanced skills like using Boolean operators, critical appraisal and finding levels of evidence they seem to require more training.

## 1. Introduction

Evidence-based practice (EBP) is defined as integrating individual clinical expertise with the best available external clinical evidence from systematic research [1]. EBP is an iterative process that includes five steps in the following order: formulating an answerable question, tracking down the best evidence, critical appraisal of the retrieved evidence, application of the appraised evidence in individual clinical practice, and finally assessing the outcome of the process and calibrating accordingly [2,3,4]. 

Searching electronic databases is an essential skill for a clinician who wishes to base his/her clinical practice on research evidence to provide high-quality service to patients seeking care [3]. This act of searching and retrieving evidence falls under the second step of the EBP process—tracking down the best evidence. From a pain management context, new research is constantly evolving, and it is very important that clinicians can formulate appropriate research questions to access the latest evidence so they will be able to provide appropriate evidence-based high-quality care [5]. Literature searching is essentially an art and a skill that improves with training and practice. 

Multiple steps are involved in the process of searching and retrieving evidence [6,7]. First and foremost, the clinician should be able to identify the clinical problem for which they need an answer. The next step is to frame a research question to clearly describe the clinical problem. The PICO (population, intervention, control, and outcomes) format is usually used and is an effective way to frame a research question [7,8,9,10]. Previous research studies have found that using the PICO format to frame research questions is an important attribute and is associated with better overall reporting quality [10,11,12,13].

The next step is to identify the sources of evidence that could provide articles that can answer this question. There are many sources of literature that can be accessed to complete these searches [6,14]. Previous studies have supported the use of electronic databases as the best source; however, there should also be an open mind to look into other sources [15]. Electronic databases are so handy because they can be searched on the go from the clinic or from home. Electronic databases allow for complicated searches and save a lot of time and resources [6]. Sometimes, clinicians’ inability to identify the correct source can leave their need for answers unmet [16]. The next step is the actual search is refining the search and retrieval of relevant articles [6,14,17]. Once articles are retrieved, the evidence should be critically appraised and then integrated into practice keeping in mind patient preferences [3,6,14,18].

Previous studies have documented the experience of clinicians using online electronic databases for literature review. Rosenbaum and colleagues have reported that many participants in the study that involved searching a commonly known electronic database displayed feelings of ineptitude, alienation, and frustration because of technical jargon and maneuvering difficulties [19]. Another study found that many clinicians found that they did not have adequate training in searching for medical literature. A recent study of nurses in Israel found that they were less likely to search professional databases for evidence-based medical information [20].

There are few studies that have been conducted to understand the experience of clinicians, using online databases to find evidence related to pain management. A previous study that looked into the knowledge, attitudes, and behaviors of 675 physicians, nurses, occupational therapists, physical therapists, and psychologists who were involved in pain management and found that all of them had similar levels of knowledge and attitudes toward EBP; however, the physicians did better with the implementation of EBP [21]. There are no studies exploring the experiences of the clinicians involved in pain management as they seek to implement EBP.

The mixed method is a design which involves the integration of qualitative and quantitative data to better understand a concept that is being studied [22]. Mixed methods is a type of research inquiry that provides a holistic approach by enabling researchers to bridge the two streams of qualitative and quantitative methodology [23]. Mixed method designs are becoming more prevalent in healthcare and health-related research [24,25]. 

The experience of clinicians, using online databases to find evidence related to pain management, is an area that has not been studied well. Getting to know their experiences can enable organizations and evidence repositories to plan better means of pushing out evidence to clinicians and modify current methods to provide a better experience of searching for evidence, thus improving evidence-based practice. We chose a mixed methods design because it can give us a greater understanding of the experience of clinicians as they use electronic databases to find evidence related to pain management. This would give us a greater insight into the matter just beyond numbers and validate the quantitative assessment with qualitative data [26]. Hence, this study will try to answer the following research question: what are the experiences of clinicians accessing electronic databases to search for evidence on pain management?

The purpose of the current study is to understand the experiences of clinicians accessing electronic databases to search for evidence on pain management using a mixed methods approach. 

## 2. Methods

### 2.1. Study Design

Mixed methods—convergent parallel strategy. This study involved two parts (a qualitative and a quantitative part) that ran parallel. The study followed the guidelines recommended by Leech [18]. We used chart stimulated recall (CSR) technique where participants were interviewed using a structured interview guide (See Appendix A), and as they were being interviewed, they would be scored on a set of pre-determined practice dimensions that are compared at baseline and 15 months. CSR interviews were completed at baseline (0 months) and in (15 months). It is scored on a 7-point Likert scale. For the current study, we are reporting the results of baseline data. Data were collected between 2013 and 2015. 

### 2.2. Participants

A purposive sample was recruited for the study. Healthcare professionals who were part of a randomized control trial looking into the effectiveness of push versus pull [19] strategies for disseminating evidence on pain management signed up for the study voluntarily by checking an option in the forms provided for the study asking them if they were interested in discussing their experiences in searching for evidence on pain. Our participants consisted of 37 healthcare professionals who are actively involved in pain management (PTs (13), OTs (14), RN (8) and Psychologists (2)). Demographic characteristics are listed in Table 1.

### 2.3. Procedure

Ethics approval for this study was obtained through Hamilton Integrated Research Ethics Board of McMaster University, Hamilton, Ontario, Canada (REB#10-586). We followed all guidelines described. The study was conducted from the MacHand Lab at McMaster University, Hamilton, Canada. Once participants were recruited, a research coordinator contacted them via mail and telephone and sorted a convenient time for a telephonic interview. The baseline interviews were completed for 37 participants over the phone. Structured interviews (*n* = 37) were continued until saturation (the point at which no new data emerged). The interview guide is attached in Appendix A. All interviews were audiotaped using an Olympus VN-3100PC recorder. The interviews were scored using a chart-stimulated recall guide using a 7-point Likert scale. (Appendix B) The participants were asked to explain two of their recent patients where pain is the major presenting complaint, and we asked them to describe them and how they managed their pain. They were then asked a series of questions about how the research would help to manage their patients’ problems, how they searched for evidence, where they searched, and did the research evidence help them inform their clinical decisions. 

The following nine dimensions were examined:Identification of an issue about the patient’s pain for clinical decision-making;Formulating the question in an answerable manner;Identification of the source of research evidence to answer this question;Knowing how to find research evidence that would answer this question;Articulate the general conclusion from relevant research;Ability to articulate specifics of dosage or expected effects from relevant research;Ability to name contraindications or considerations derived from relevant research;Ability to differentiate high-quality versus low-quality studies;Ability to identify and cite systematic reviews pertaining to this question.

### 2.4. Data Analysis

#### 2.4.1. Qualitative

All interviews were transcribed verbatim and then checked by the interviewer for accuracy. For the analysis, we used a general descriptive qualitative search methodology, as described by Sandelowski [20]. NVivo 10 (QSR International, Chadstone, Australia) qualitative data management software was used for data organization. Data collection and analysis occurred following an inductive, iterative process. Coding was framed in three progressive stages: open, axial, and selective. Open coding consisted of a line-by-line analysis of the transcript to determine codes. As each new transcript was analyzed, data were compared with existing codes, and either an existing code or a new code was created using NVivo. This stage of analysis also involved writing reflective memos that helped later in the analysis stages. Axial coding is the second step, in which codes were compared with each other and reflective memos to form categories, representing similar codes brought together through the relating of concepts inherent in the codes. Selective coding is the last step where categories were examined and compared to each other to develop themes. The credibility and trustworthiness of the study processes were enhanced by the following means:Prolonged engagement with data in person and frequent listening to the interviews and checking if it matched the CSR scoring [21]).Verbatim transcription of interviews and member checking with the other author and peer debriefing [22].Development and maintenance of audit trail throughout the research process, to ensure that the same questions were asked with every participant process [21].Analysis of transcribed data with the other authors and making separate codes and discrepancies found were checked as well followed by team discussions at all stages [20].

#### 2.4.2. Quantitative Stream

Descriptive statistics including the mean and standard deviation for individual dimensions of the CSR scoring and overall score were calculated using SPSS version 22 (IBM corporation, New York, NY, USA).

## 3. Results

### 3.1. Quantitative

Participants were scored on nine practice dimensions as they were being interviewed. The mean total CSR score was 35.62 out of 63 (56.53%). This indicated that the participants had a low to moderate level of knowledge of the literature searches. When the individual dimensions were explored, it was found that the participants scored better on basic evidence search skills (mean scores > 4), but when it came to advanced skills such as appraising evidence or grading quality of evidence, etc., they had lower scores (mean scores < 4) (See Table 2).

### 3.2. Qualitative

Seven main themes were identified based on the analysis of the qualitative interviews. 

#### 3.2.1. Theme 1: Formulating a Research Question

The use of the PICO format came up during the interviews. Two out of the thirty-seven participants used the PICO format as a guideline to formulate their research questions.


*“I guess what we learned in school was PICO, population, intervention, comparison or outcome so, for both of those, I think I did more population so low back pain.”*
(OT 643).

This qualitative finding is backed by the moderate CSR score of 4/7 with a range of 2 to 7 for the dimension of “able to formulate a research question”.

While exploring the low utilization of the PICO format to formulate research questions, participants said that not all searches are based on interventions, so they had to modify their search terms. This is what another participant mentioned in this regard; *“I try to use the PICO format sometimes……”* (PT 745).

#### 3.2.2. Theme 2: Sources of Evidence Accessed

During the interviews, the participants reported searching for and accessing evidence in various electronic databases. PubMed [*n* = 13 (35.1%)] was the most commonly used followed byCINAHL [*n* = 9 (24.3%)], MEDLINE [*n* = 8 (21.6%)], Cochrane [*n* = 8 (21.6%)], Google Scholar [*n* = 6 (16.2%)], and PAIN+ [*n* = 6 (16.2%)]. The others that were used are PsychInfo [*n* = 5 (13.5%)], OVID [*n* = 3 (8.1%)], and EMBASE [*n* = 2 (5.4%)]. The following extracts describe this finding.


*“I would usually, where I would usually go sort of these two either go to PubMed or Google Scholar.”*
(OT 632).


*“I use the CINAHL database, OVID, Google Scholar, trying to find some journals that lend credibility.”*
(OT 639).

Rehab participants tend to search databases specific to their profession, the American Physical Therapy Association (APTA), Rehab+, the Canadian Journal of Occupational Therapy, and the American Journal of Occupational Therapy, while RNs and psychologists were interested in Psych Info, the clinical information access portal, and the Journal of the international association for the Study of Pain.


*“For me, I always go to the APTA, … I might look in journals, I read abstracts to articles, maybe pain medication articles, the Journal of Pain and Orthopedic.”*
(PT 651).

Clinicians use a variety of search engines but were able to tap into the most common sources of evidence. This qualitative finding is backed by the fact that the clinicians scored high (Mean 5.7/7 (Range 1–7)) on the CSR score for the dimension “Could identify the source of research evidence to answer this question”.

#### 3.2.3. Theme 3: Search Strategy

Participants described a variety of search strategies that they used to search the electronic databases. When using search terms, participants in our study tend to use different forms of the same word to be more comprehensive in their search. 


*“…if there were 2 ways of saying something, sometimes I would use ‘OR’, both back pain or chronic back pain or back pain in search engines.”*
(OT 643).

Only a minority of participants reported the use of Boolean operators to connect search terms came up during the interviews. Three out of the thirty-seven participants used them for their searches. The three participants who used these Boolean operators were able to describe how they would use them in a typical search. They indicated that they use plus sign “+” to connect search terms.


*“Like what I try and do, for example, the first one, I would search diabetes + insulin + chronic pain and that’s how I do that one.”*
(OT 639).

During the interview the reasons for using “AND” during the search which is for finding synonyms or for increasing the specificity of the search also came up. 


*“I would probably put chronic pain AND whiplash AND female AND Interventions as well. I know it’s a really long search but I would probably add interventions afterwards.”*
(OT 649).

The above qualitative findings of a relatively small number of participants using Boolean operators to enhance their search and general ability to use different forms of the same word is reflected in the moderate CSR score (Mean 4.2/7 (Range 1–7)) for the dimension of “Could identify how to find research evidence that would answer this question”.

#### 3.2.4. Theme 4: Refining the Yield

The participants in our study used different methods to narrow down or refine the search. Some use title scan, abstract scan, and full-text download. 


*“Usually I would scan the titles… if the title looks like something that would actually be relevant, I would read the abstract. If the abstract looks good I would actually download the article… then actually reading the articles.”*
(OT 643).


*“Probably eliminate some by the titles and then I would look at when you read the article they have like an abstract.”*
(OT 643).

Some limited the search by the date of publication, by looking for recent articles.


*“Usually I would scan… the year of publication just because if it’s super outdated, I probably wouldn’t look at it.”*
(OT 643).

Another way was they looked into relevant articles section of the databases or/and the articles that cited the current paper.


*“……can do related articles or cited by articles so you can ballpark a whole new list of 30 articles that you get more recent ones.”*
(OT 643).

Some just read the article that came at the top of the list.


*“Logistically I would look at the ones that came up first, I would open them, I guess I might be pulled by certain titles, I would.”*
(OT 643).

#### 3.2.5. Theme 5: Barriers and Facilitators 

A theme that came up during the interviews was barriers and facilitators for literature searches. Access, time, and resources were noted as major barriers, while institutional access was a major facilitator. The barriers and facilitators are summed up in Table 3 with illustrative quotes.

All these barriers and facilitators lead to our next theme on the sources that clinicians use for clinical decision-making. These barriers not only lead to the next theme but also explain part of the reason why clinicians do what they do with the sources they seek to aid their clinical decision-making.

#### 3.2.6. Theme 6 Clinical Decision Making 

For making clinical decisions, the participants in our study often looked up to their peers and were influenced by their knowledge of their peers within their field, and their professional expertise was valued.


*“…by talking to a physiotherapy colleague about that…So I didn’t recommend the back braces.”*
(OT 643).

Some participants indicated that they give greater priority to clinical experience gained over the years dealing with various kinds of patients, in decision-making. 


*“I was a registered nurse for 20 years and I’ve been a nurse practitioner since May, just knowing that they were contraindications. it’s basically from your experience…”*
(RN 750).


*“I have 25 years’ experience so here it comes; I give a set of exercises that have been effective for me and I use them. Have they been well researched, I doubt it…”*
(PT 653).


*“Well you know that’s probably just experience that I’ve learned over the years”*
(PT 673).

Some participants relied on the entry-level training and stuck on to basic training and followed even after 20 years of experience.


*“ ……and it’s something that learned in school?”*
(RN 750).


*“… I learnt this in school when I did my undergrad.”*
(PT 653).

Some participants also looked up actual evidence that would support their decisions.


*“Pretty much it’s evidence-based as far as the medications utilization in a particular patient group and it’s always tried to get informed by the evidence notes if it’s available.”*
(RN 677).

One participant said that decisions are made both on experience and research and a little bit of both.


*“I’ve learned through colleagues, a little bit through research and upper management as well as too, and training as well as to how often they should be performing relaxation, how often they should be planning out their day and that kind of stuff, it’s been more on the job training I guess you could say.”*
(OT 649).

The participants scored low on the three dimensions that explored their ability to extract results from the research article, interpret them, and apply them to their clinical question at hand. The dimensions are to articulate the general conclusion from relevant research (Mean 4.1/7 (Range 1–6)) the ability to articulate specifics of dosage or expected effects from relevant research (Mean 3.9/7 (Range 2–7)) and the ability to name contraindications or considerations derived from relevant research (Mean 3.6/7 (Range 1–7)).

The low CSR scores for these dimensions are explained by the qualitative findings of most clinicians, looking on to peers, experience, and entry-level training to make clinical decisions rather than looking into the actual scientific evidence and interpreting them to answer clinical questions related to patient care.

#### 3.2.7. Theme 7: Knowledge and Awareness about Appraising the Quality of Evidence

Most participants had limited knowledge of the types of research designs, levels of evidence, and how to apprise an article for their quality. Some participants were aware of the two types of research—quantitative and qualitative. 


*“… there are various kinds of study designs, there’s quantitative … and then there’s qualitative.”*
(PT 673).

Fewer participants were able to talk about the levels of evidence.


*“Yes, the National Guidelines for Acute Pain Management in Australia, there is a chapter on it, but I haven’t read it in many years, but it talks about level 1, level 2, level 3, level 4, and I think it goes up to 5; Level 5 is the more from experience by case studies, I think level 1 is randomized control trials, good research techniques.”*
(RN 733).

When asked to define and describe a systematic review, only a few were able to do this.


*“…………it has been very high-quality research that’s been done on this and there has been good systematic reviews and meta-analyses and evidence-based guidelines.”*
(RN 754)

The lack of advanced skills to understand the levels of evidence and the ability to appraise the quality of evidence in our participants was reflected in the low CSR score (Mean 2.4/7 (Range 1–7)) for the dimension of “Was able to identify and cite systematic reviews pertaining to this question” and (Mean 3.03/7 (Range 1–7)) for the dimension of “Was able to differentiate high-quality versus low-quality studies”. 

## 4. Discussion

This study explored the retrospective recall of the experiences of clinicians using electronic databases to search for evidence related to pain. We used a mixed methods approach and found that the quantitative data backs up the qualitative data. The themes ranged from searching for evidence, refining the yield, barriers encountered during the search, and quality of studies searched. We found that clinicians performed well with their basic evidence search skills such as framing research questions, searching appropriate databases, using correct search strategies, etc. However, when it came to the appraisal of the literature and use of evidence in clinical decision making there were mixed responses.

Formatting a well-built clinical question (foreground question) is the fundamental skill in an evidence-based search strategy, and these questions should be relevant to client problems and should be able to direct the search to get precise answers [5,6,7]. Despite PICO being prescribed to enhance the literature search strategies to help ensure a comprehensive and exhaustive literature review [27] and being also endorsed by the Cochrane Collaboration as a model for developing review questions [28], only 2 out of the 37 participants responded as using the PICO format to frame research questions. A similar trend has been reported in previous studies. A systematic review of 313 articles published in anesthesiology journals found that 96% did not apply PICO format to their research question [10]. The reasons that we found for this behavior from the interviews were that they had forgotten about the PICO format as they had learnt it at school a long time ago or they were not aware of the PICO format.

In the current study we found that most of the clinicians used PubMed, Medline, Cinhal, and cochrane followed by Google Scholar. This was no surprise as this trend is evident in previous studies that have found PubMed and Google Scholar to be the most accessed electronic databases by health professionals [29,30,31]. It should be noted that in spite of Google Scholar being one of the popular search engines used by healthcare professionals, the relatively low quality of the metadata and the difficulty to extract that metadata make it challenging to use Google Scholar data in bibliometric analyses [32]. In our study, some of the participants used a combination of search engines while most of them did not. This puts the clinicians at risk of losing the best evidence available. It is highly recommended that clinicians use more than one electronic database to perform a comprehensive search of evidence [33]. 

Another trend that was observed was that clinicians in a particular profession tend to use electronic databases and online portals that are dedicated to their profession. For example, nurses and psychologists tend to use CINAHL and Psychinfo more than other databases as they contained more articles that were related to their field. This behavior is well documented in earlier studies. A recent study has reported that nurses commonly use MEDLINE, CINAHL, and Scopus to conduct literature searches [34]. Clarke and colleagues [35] in their literature review to understand the information needs of physicians and nurses have found that they accessed job-specific resources to find practitioner-oriented information. Bolstrom and colleagues [36] have reported seeking research that is related to clinical practice as a major predictor of research utilization (OR = 5.56, *p* = 0.019). It is highly recommended that clinicians use electronic databases recommended for their discipline and at the same time use multiple electronic databases [33].

Boolean operators help to translate a clinical question to a specified format which the search engine can understand. This would help to filter off content that is irrelevant to the research question. However, in our study, only 2 out of the 37 participants used Boolean operators routinely in their searches. The reasons for this could be that many participants do not have the knowledge of these advanced literature search skills and these skills are not consistently taught at schools that train healthcare professionals. It may also be that clinicians may think that it is complicated and time-consuming to use these Boolean operators, which in reality would save their time by avoiding irrelevant articles. It has been previously reported that the skills required to acquire, appraise, and integrate knowledge into clinical practice are a major determinant of EBP [36,37]. Better continuing education opportunities around the area of literature searching skills would help clinicians understand the usefulness of these operators and implement them in everyday search from evidence.

The participants in our study also talked about various barriers and facilitators of online search for pain evidence. The most important barrier was access to the databases. Similar reports have emerged in the literature, and our findings confirm them [36,38]. Bostrom and colleagues [36] found that access to databases was a major predictor of research utilization (OR = 6.65; *p* < 0.005). This calls in for changes in policy that would enable increased access to electronic databases at workplaces. Additionally, another important barrier was lack of time which has become an endemic problem for research utilization. Previous research studies have identified this issue, and our qualitative finding adds credibility to the quantitative findings of those studies [21,39]. 

In our study, we found that the participants in our study scored less in the CSR (<4) for this area. Additionally, most of the participants in the current study were not able to identify the levels of evidence. As the clinical literature grows in a rapid pace, it becomes essential so that EBP becomes doable so as to improve patient care. It is important that clinicians would be able to differentiate between high quality and low quality evidence. Inability to identify the levels of evidence and classify articles of high and low methodological quality would impede the implementation of EBP in clinical practice resulting in patient care that is not current. 

The participants in the current study reported that they looked up to their peers to help with clinical decision-making. This has been previously reported and is more prevalent [35]. The reasons for this could be the quickness and the ease of getting information from peers [35]. However, the pieces of information gathered this way might be outdated and sometimes incorrect, affecting the quality of care rendered to patients. Additionally, another source was a previous experience; previous experience is good in that it makes a person expert, but when it is not combined with current evidence then it makes it outdated and may hamper evidence-based practice. Studies suggest that 30 to 40% of patients do not receive care according to current scientific evidence [40]. Current models of patient care focus on evidence-based practice, and it is important that clinicians move from traditional knowledge sources to evidence-based practice to enhance the quality of the care delivered.

To improve the implementation rates of EBP, it might be necessary to conduct refresher courses on literature searches for practicing clinicians and make it mandatory in students in professional schools for healthcare practice. A recent international, multi-disciplinary, cross-sectional study of pain knowledge and attitudes in nursing, midwifery, and allied health professions students found that there was little difference in pain knowledge and attitudes between all first- and final-year students other than physiotherapy [41]. This indicates the need to incorporate evidence-based pain management education in the curriculum during training. We also recommend an e-learning approach as a way of providing continuing professional development especially when it comes to learning about literature searching and critically appraising the evidence to assist with providing evidence-based care. A systematic review of the effectiveness of e-learning has found that e-learning has become an accepted tool or approach among medical, nursing, and allied healthcare professionals for continuing medical education [42]. This is because contemporary information can be delivered rapidly and flexibly, adopting varying formats, while the learner can go at their desired pace [42,43].

The strengths of the current study are as follows: We used a mixed methods approach of integrating qualitative and quantitative methods. We used the quantitative data that was collected to back up the qualitative interview data. This enabled us to better explore the dimensions of evidence search that were studied. We used the chart simulated recall (CSR) technique to collect our quantitative data. This is valid and reliable for professionals such as physicians, nurses, and rehabilitation professionals to collect quantitative data in mixed methods studies and has added to the credibility of the data collected in this study. This study included participants from different geographical regions (Canada, United States, Australia, New Zealand, and South Africa) thus giving a snapshot of how evidence-based practice is implemented in different parts of the globe. Interpretation and findings of the study have few limitations. The number of professional were not equal, so we did not try to compare the differences across professionals. Our study also had some limitations. We included a purposive sampling from an RCT in which not all professions were equally represented in this study. This may affect the generalizability of the study findings. Even though there were only two psychologists participating in the study, their transcripts were also analyzed and included since their behaviors are relatively less studied when compared with other professionals. Only nurses, physios, OTs, and psychologists were included in this study, which may affect the generalizability of these findings to other professions. 

In conclusion, using a mixed methods approach, we found that clinicians were performing well with their basic literature review skills, but when it came to advanced skills like using Boolean operators, critical appraisal, and finding levels of evidence they seem to require more training. 

## Figures and Tables

**Table 1 healthcare-11-01728-t001:** Demographic data.

Characteristics	*n* = 37
Age	*n* (%)
20–35	10 (26.4)
36–45	9 (23.4)
46–55	12 (33.0)
56+	6 (17.2)
Clinical Designation	*n* (%)
OT	14 (37.8)
PT	13 (35.1)
RN	8 (21.6)
Psychologist	2 (5.4)
Received advanced clinical certifications	16 (43.2)
Years of clinical training	*n* (%)
Less than 2 years	13 (35.1)
2–5 years	16 (43.2)
Above 5 years	8 (21.6)
Location of practice	*n* (%)
Urban	25 (67.6)
Rural	7 (18.9)
Both	5 (13.5)

**Table 2 healthcare-11-01728-t002:** CSR scores.

Dimensions (*n* = 37)	Min	Max	Mean	SD
1. Identifies an issue about the patient’s pain for clinical decision-making	2	7	4.73	1.19
2. Is able to formulate the question in an answerable manner	2	7	4.08	0.75
3. Could identify the source of research evidence to answer this question	1	7	5.70	1.95
4. Could identify how to find research evidence that would answer this question	1	7	4.20	1.49
5. Was able to articulate the general conclusion from relevant research	1	6	4.10	1.18
6. Was able to articulate specifics of dosage or expected effects from relevant research	2	7	3.90	1.16
7. Was able to name contraindications or considerations derived from relevant research	1	7	3.60	1.14
8. Was able to differentiate high-quality versus low-quality studies	1	7	3.03	1.61
9. Was able to identify and site systematic reviews pertaining to this question	1	7	2.41	1.80

**Table 3 healthcare-11-01728-t003:** Barriers and facilitators with illustrative quotes.

Barriers	Illustrative Quotes
Limited/no access to databases	*“Right now, I don’t have access to, … to all the databases. Right now, I don’t go through my workplace. So, I actually generally just start with just a Google Scholar search, see if I can find something there.”* (OT 683)
Limited time and access	*“I would like to be searching journals. But one of the limitations is time and the other is access, our hospital from a rehab perspective doesn’t have access to any journals. So we’re limited to what you can access.”* (OT 679).
Limited Staffing	*Well, how I actually do it we have, our hospital has very limited staff resources also very limited time for research.” (* *OT 679)*
Difficulty in navigation	*“Medline search or PubMed. I‘ve looked at them but honestly I don t find them very useful, I find them difficult to navigate…”* (PT 651).
Facilitators	Illustrative quotes
Access to databases	*“I would log on to, I have access to the University of Western Ontario library, so I would log on to…”* (PT 663).

## Data Availability

Not applicable.

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
