# Peer review of "Understanding the Experiences of Clinicians Accessing Electronic Databases to Search for Evidence on Pain Management Using a Mixed Methods Approach"

_healthcare, 2023, doi:10.3390/healthcare11121728_

Round 1

Reviewer 1 Report

Thank you very much for the chance of reviewing the manuscript about using the capability of clinicians to access electronic databases to search for evidence for pain management. This study adopted several methods to conduct data analysis. Some information needs to be added to emphases the meaningfulness of this study. The comments are as follows.

In the introduction, I think more can be added on how the importance of using electronic databases for managing pain among clinicians. Also, the significance of this study can be mentioned as the contribution of this study.

There is a section about the purpose of this study. You mentioned a number of dimensions; however, this part should be more focused on the purposes of this study instead of the dimensions. Or you can think about how to make the dimensions become the purpose or the meaningfulness of this study.

What are the rationales to recruit this sample size? Do you have any representative studies to support it?

For the findings and discussions, any different opinions among different professionals? I think these findings will be meaningful because accessing the electronic databases among different professionals may be critical to managing pain.

Author Response

   Response to reviewers’ comments

 Reviewer#1

The authors would like to thank the reviewer for the time spent in reviewing this manuscript and great feedback provided. I am sure this has helped this manuscript take a better shape. Please find our point-by-point response for your feedback.

Thank you very much for the chance of reviewing the manuscript about using the capability of clinicians to access electronic databases to search for evidence for pain management. This study adopted several methods to conduct data analysis. Some information needs to be added to emphases the meaningfulness of this study. The comments are as follows.

  • In the introduction, I think more can be added on how the importance of using electronic databases for managing pain among clinicians. Also, the significance of this study can be mentioned as the contribution of this study.

Response: We have now included some more information on the importance of using electronic databases.

  • There is a section about the purpose of this study. You mentioned a number of dimensions; however, this part should be more focused on the purposes of this study instead of the dimensions. Or you can think about how to make the dimensions become the purpose or the meaningfulness of this study.

Response: We have now moved the dimensions to the methods section of the study. .

  • What are the rationales to recruit this sample size? Do you have any representative studies to support it?

Response: This is a purposive sample as participants from a larger trial were asked to sign up for this study and the number participants who signed up for this study were included.

  • For the findings and discussions, any different opinions among different professionals? I think these findings will be meaningful because accessing the electronic databases among different professionals may be critical to managing pain.

Response: We have now described them in the results and discussion. The biggest one was the different types of databases they use.

Reviewer 2 Report

The article is well-written, and organized, and contributes significantly to the literature on using electronic databases in evidence-based clinical practice in pain management. The study used a mixed research approach to explore health professionals' experiences searching for pain-related evidence, and identifying both strengths and gaps in their skills and knowledge. The article presents a comprehensive review of the relevant literature, uses rigorous research methods, and provides a clear and well-reasoned discussion of the findings. However, there are some issues that need to be reviewed:

Abstract

The abstract could provide more information about the results found in each of the seven competencies assessed and how they were integrated into the analysis. In addition, it would be useful to include information about the main conclusions of the study and their relevance to clinical practice.

Introduction:

The authors used a structured approach to write the introduction, initially providing the definition of evidence-based practice (EBP) and its importance in healthcare. They then discussed the five-step process involved in EBP and the challenges medical professionals face in finding evidence for pain management.

The authors stated that PBE is considered the current best practice in any health and health-related profession. This statement may be mistaken. Not all health professionals can follow this practice. Do not generalize and state that PBE is a widely accepted practice in many health professions.

The introduction presents a good conceptualization of EBS and the search in the literature. However, it is necessary to reinforce the research question that the study seeks to answer. Even though the authors cite previous studies, they have not critically explored the strengths and limitations of these studies.

The last few paragraphs explain the use of mixed methods research to investigate physicians' experiences in using electronic databases to find evidence related to pain management. I suggest that the authors explicitly state the research question and be more critical in evaluating previous research.

Objective and hypotheses:

The aim of the study is well defined in the introduction and the nine hypotheses are related to EBS, discussed, and conceptualized in the introduction.

Method:

The authors used the mixed sequential explanatory approach method, integrating qualitative and quantitative data. The use of the approach is appropriate because the authors already had an initial hypothesis but wished to explore it in greater depth. The qualitative analysis was conducted through individual interviews with participants, while the quantitative analysis was conducted using Chart Stimulated Recall (CSR), a technique that involves using a visual tool to help participants recall and describe their evidence-seeking processes. The interviews were transcribed verbatim and analyzed using a general descriptive qualitative research methodology. The authors used NVivo 10 software for data organization.

Some points need to be specified for better internal and external validity of the study:

The sample was recruited through an option on a form for a different study, which may have led to a biased sample, and was recruited from a randomized control study, which may limit the generalizability of the results to other contexts.

The CSR technique is an important and appropriate approach to exploring experiences. The questionnaire used a 7-point Likert scale to score the interviews may have privacy in capturing nuances and more subtle details of the data.

Although the qualitative analysis followed a well-established approach (general descriptive analysis), the selection of codes and categories is subjective and may be subject to the personal views of the researcher when not using an appropriate theoretical model.

An overview of CSR scores was provided but it was not mentioned which analyses were performed (descriptive/inferential), making it difficult to identify more complex patterns or correlations in the data.

Because the study focused on a specific group of healthcare professionals, the findings may have limitations in applicability to other clinical or patient areas.

After data collection, the quantitative and qualitative results were compared and integrated with a cross-analysis. The meta-analysis was performed through the thematic analysis of the qualitative and quantitative data, but the triangulation did not clearly present how this crossover was performed, which limits the validation of the results and a more comprehensive and holistic view of the research topic. The importance of identifying constraints and areas for improvement as opportunities for improvement for future studies is emphasized.

Results:

The results presented are consistent with the methodology employed. The qualitative analysis emerged seven themes related to formulating a research question, sources of evidence accessed, search strategy, and refinement of the output. However, some issues need to be further clarified in the text:

Only two of the 37 participants reported using the PICO format, which is widely recommended for formulating clinical research questions and is a useful tool to help researchers identify relevant search terms for formulating research questions. This may be an important limitation. The same refers to the use of Boolean operators, which was indicated to be used by only three of these respondents, which can be attributed to low scores in advanced research skills.

The sources of evidence were also limited, in which the use of databases from the specific area of the professional was specified. The participants even mentioned that there are access limitations to these databases, but the authors did not make clearer inferences regarding the themes and the interviewees' statements, which could facilitate the reader's understanding.

Discussion and Conclusion:

The discussion does not provide a comprehensive review of the literature related to electronic database searching and evidence-based practice, considering the importance of the results that were presented.

Some of the limitations of the study were not addressed, such as the small sample size of psychologists and the fact that participants were not evenly distributed across professions. which affects the power of generalizing the results to other settings.

While it points to the importance of continuing education opportunities for health professionals in literature search skills, it does not provide specific recommendations on how this might be implemented or what types of training would be most effective.

The internal and external validity of the study:

The strengths of internal validity refer to the mixed data approach and the inclusion of participants from different fields, as well as, the use of the simulated chart recording (SSR) technique and data triangulation, which raise the internal validity. As for external validity, the study is limited to the sample, which was taken from a group that participated in a randomized clinical trial, was small and not equally distributed among professions, which reduces the power of generalization.

Author Response

   Response to reviewers’ comments

 The authors would like to thank the reviewer for the time spent in reviewing this manuscript and great feedback provided. I am sure this has helped this manuscript take a better shape. Please find our point-by-point response for your feedback.

Reviewer#2

The authors would like to thank the reviewer for the time spent in reviewing this manuscript and great feedback provided. I am sure this has helped this manuscript take a better shape. Please find our point-by-point response for your feedback.

The article is well-written, and organized, and contributes significantly to the literature on using electronic databases in evidence-based clinical practice in pain management. The study used a mixed research approach to explore health professionals' experiences searching for pain-related evidence, and identifying both strengths and gaps in their skills and knowledge. The article presents a comprehensive review of the relevant literature, uses rigorous research methods, and provides a clear and well-reasoned discussion of the findings. However, there are some issues that need to be reviewed:

Abstract: The abstract could provide more information about the results found in each of the seven competencies assessed and how they were integrated into the analysis. In    addition, it would be useful to include information about the main conclusions of the study and their relevance to clinical practice.

 Response: We have included some implications in the abstract, however due to journal restrictions on the length of the abstract more information about the results found in each of the seven competencies assessed and how they were integrated into the analysis.

Introduction:

  • The authors used a structured approach to write the introduction, initially providing the definition of evidence-based practice (EBP) and its importance in healthcare. They then discussed the five-step process involved in EBP and the challenges medical professionals face in finding evidence for pain management.The authors stated that PBE is considered the current best practice in any health and health-related profession. This statement may be mistaken. Not all health professionals can follow this practice. Do not generalize and state that PBE is a widely accepted practice in many health professions.

Response: We have removed this sentence

  • The introduction presents a good conceptualization of EBS and the search in the literature. However, it is necessary to reinforce the research question that the study seeks to answer. Even though the authors cite previous studies, they have not critically explored the strengths and limitations of these studies. The last few paragraphs explain the use of mixed methods research to investigate physicians' experiences in using electronic databases to find evidence related to pain management. I suggest that the authors explicitly state the research question and be more critical in evaluating previous research.

Response: We have worked on the introduction to include more research and also to explore the strengths and weakness of the previous literature. Also we have clearly stated the research question.

 Objective and hypotheses:

  • The aim of the study is well defined in the introduction and the nine hypotheses are related to EBS, discussed, and conceptualized in the introduction.

Response: thank you for this feedback.

  • Method: The authors used the mixed sequential explanatory approach method, integrating qualitative and quantitative data. The use of the approach is appropriate because the authors already had an initial hypothesis but wished to explore it in greater depth. The qualitative analysis was conducted through individual interviews with participants, while the quantitative analysis was conducted using Chart Stimulated Recall (CSR), a technique that involves using a visual tool to help participants recall and describe their evidence-seeking processes. The interviews were transcribed verbatim and analyzed using a general descriptive qualitative research methodology. The authors used NVivo 10 software for data organization. Some points need to be specified for better internal and external validity of the study: The sample was recruited through an option on a form for a different study, which may have led to a biased sample, and was recruited from a randomized control study, which may limit the generalizability of the results to other contexts.

Response: we have included this as a limitation in our discussion now.

  • The CSR technique is an important and appropriate approach to exploring experiences. The questionnaire used a 7-point Likert scale to score the interviews may have privacy in capturing nuances and more subtle details of the data. Although the qualitative analysis followed a well-established approach (general descriptive analysis), the selection of codes and categories is subjective and may be subject to the personal views of the researcher when not using an appropriate theoretical model. An overview of CSR scores was provided but it was not mentioned which analyses were performed (descriptive/inferential), making it difficult to identify more complex patterns or correlations in the data.

Response: A descriptive analysis of the CSR score was completed. We have now clearly mentioned this in the methods section

  • Because the study focused on a specific group of healthcare professionals, the findings may have limitations in applicability to other clinical or patient areas.

Response: This is now included as a limitation

  • After data collection, the quantitative and qualitative results were compared and integrated with a cross-analysis. The meta-analysis was performed through the thematic analysis of the qualitative and quantitative data, but the triangulation did not clearly present how this crossover was performed, which limits the validation of the results and a more comprehensive and holistic view of the research topic. The importance of identifying constraints and areas for improvement as opportunities for improvement for future studies is emphasized.

            Response: We have now integrated the quantitative data into the qualitative findings and how       one explains the other.

  • Results: The results presented are consistent with the methodology employed. The qualitative analysis emerged seven themes related to formulating a research question, sources of evidence accessed, search strategy, and refinement of the output. However, some issues need to be further clarified in the text:

Only two of the 37 participants reported using the PICO format, which is widely recommended for formulating clinical research questions and is a useful tool to help researchers identify relevant search terms for formulating research questions. This may be an important limitation. The same refers to the use of Boolean operators, which was indicated to be used by only three of these respondents, which can be attributed to low scores in advanced research skills.

The sources of evidence were also limited, in which the use of databases from the specific area of the professional was specified. The participants even mentioned that there are access limitations to these databases, but the authors did not make clearer inferences regarding the themes and the interviewees' statements, which could facilitate the reader's understanding.

Response: These are now included in the results and discussion section.

Discussion and Conclusion:

  1. The discussion does not provide a comprehensive review of the literature related to electronic database searching and evidence-based practice, considering the importance of the results that were presented.

Some of the limitations of the study were not addressed, such as the small sample size of psychologists and the fact that participants were not evenly distributed across professions. which affects the power of generalizing the results to other settings.

Response: These limitations are acknowledged now

  1. While it points to the importance of continuing education opportunities for health professionals in literature search skills, it does not provide specific recommendations on how this might be implemented or what types of training would be most effective.

Response: We have included the specifics in a new paragraph now.

The internal and external validity of the study:

  1. The strengths of internal validity refer to the mixed data approach and the inclusion of participants from different fields, as well as, the use of the simulated chart recording (SSR) technique and data triangulation, which raise the internal validity. As for external validity, the study is limited to the sample, which was taken from a group that participated in a randomized clinical trial, was small and not equally distributed among professions, which reduces the power of generalization.

Response: This is now discussed in the limitations.

Reviewer 3 Report

Thank you for the opportunity to review your paper. It’s an important area and you have adopted an interesting approach. However, I think there are issues with the manuscript as it stands.

I appreciate that this will probably be disheartening but it’s important for research to be reported as well as possible.

You might find it helpful to look at the Good Reporting of A Mixed Methods Study (GRAMMS) checklist. This is an old paper but it’s accessible and should indicate how you could improve the integration of data 

and consideration of the limitations (see, O'Cathain A, Murphy E, Nicholl J. The quality of mixed methods studies in health services research. J Health Serv Res Policy. 2008;13: 92-98).

The main issues include the following (further details are provided later in my comments).

1.     There are a lot of typographical and/or grammatical errors that need addressing

2.     The references are not contemporary, and although I appreciate some references are seminal and stand the test of time, there is only one reference as far as I can see that is 2020 or later,  7 date from 1999 or earlier and most are from the early 2010s.

3.     It’s not clear when the data were collected.

4.     There is no evidence of data synthesis of the qualitative and quantitative data as would be expected in a mixed-methods study

5.     The qualitative findings have the potential to be interesting but the way they are presented, minimal text from the authors and lots of quotes is not what is expected of high-quality qualitative presentation. This section would benefit from editing and integrating the quotes more coherently.

6.     There are gaps in the paper that need addressing so that the reader can follow things through more easily.

Further comments

·       L94, when were the data collected?

·       L 101 Need to add in a time point …. CSR interviews were completed at 101 baseline (0 months) and xxxxx (15 months)

·       L105 change are to were….. Health care professionals who were..

·       Table 1. Give numbers as well as % for Age, clinical designation,  received advanced clinical certif. years of clinical training

·       L115 - approval number should be included.

·       L119 change sort to sorted.

·       l122, lower case for Guide.

·       l124, remove ‘about’.

·       l126… change start of sentence to maybe….. They were then asked a series of questions about how research would help managing their patients’ problem

·       l127 Note you need an apostrophe on patients’

·       L157 - pleases comments about reformatting table I have made in relation to Table 3.

·       L76 and elsewhere. I’d suggest putting the participant label within brackets…  e.g.,  …….and then the comparison group” (PT 745). You also need a full stop at the end of sentence/quote.

·       L182 and elsewhere - some labels are not linked to the type of participant, just a participant number. This needs to be addressed throughout.

·       Generally - it’s really hard to read the paper as the quotes just look like usual text. Something needs to be done (e.g., indenting quotes.

·       Also, I think there are many quotes where further condensing could happen. Typically you can weave short sections of quotes into text sentences rather than just describing something and then following it with a quote.

·       L167-8 delete ‘(population or patient, intervention or indicator, comparison or control, outcome)’ as you have previously provided this information.

·       L180-181. Suggest you include numbers to support statements e.g. PubMed (n=x, %) was the most commonly used followed by Google Scholar (n=x, %). The other that were used are CINAHL (n=x, %), OVID (n=x, %), Psych Info (n=x, %).

·       L181 and elsewhere you write Psych info, but should Info be capitalised

·       L194 what is -651?

·       Figure 1 is fairly pointless - it’s hard to read and isn’t particularly helpful. I would delete this and consider a different way of presenting data. Maybe a table that is in supplementary data.

·       L200-201. The first sentence needs rephrasing.

·       L243. I’m not convinced you need a table for this. I think it could be easily summed up in text. BUT if you retain the table, I’d suggest reworking it so it becomes the same width as the text to make the table look professional in the final version.

·       L252 and elsewhere - you make one point and then follow it with 3 quotes, this is not the best approach to qualitative reporting, you need to be selective in choice of ‘indented’ quotes and use the text to make points. Otherwise the reader is just reading a ’shopping list’ of quotes.

·       L252 they not thy

·       L260 - is this a sub-sub heading?

·       L268 some quotes/sentences don’t make sense, e.g.. “Being that I have been a member of New Zealand Pain Society so frequently giving up dates theories different online”.

·       L282 Delete ‘A’

·       L292 revise to    The mean total CSR score was 36 out of 52 (69.23%).

·       L292 - is there more you can present in terms of the quantitative results. The results are confusing and the reader needs to understand what you’re presenting…. You talk of the mean total score being 36 and then go on to talk of mean scores of 4-6 and 2-4… is this for individual items? I appreciate you have included a table earlier on but explain what you mean by low and well.

·       L295-6 IF you retain the content of this sentence, then revise punctuation and grammar etc to “……were explored, it was found that the participants scored better on basic evidence search skills (mean scores 4 to 6) but when it came to advanced skills such as appraising evidence or grading quality of evidence etc. they had lower scores (mean scores 2-4).

·       L295-6 ALSO - how come they could score 4 in both low and higher scores?

·       L296 here and in discussion, if you have used a true mixed methods approach you would have a section on data synthesis but it appears that the qualitative and quantitative data are just addressed separately.

·       L299 This is retrospective recall of experiences………

·       L319 and elsewhere…. Capitalise S in Google Scholar

·       L353 BUT as I’ve pointed out earlier on you have participants scoring 4 in both higher and lower categories.

·       L372 - The claim that this is mixed methods (convergent parallel strategy - see l96) can’t be a strength as there is no evidence of data synthesis.

·       L377- all references should be within the same set of brackets.

·       ·       The Discussion is inadequate at the moment.  The main issue is that it is relying on old references and not making any comment about this. If the point is that findings in this study are similar to those from years ago, this is an important point. BUT there should be more contemporary literature to draw on and to contextualise. At the moment it feels somewhat confused and it’s difficult to determine if it could make a contribution.

Author Response

   Response to reviewers’ comments

The authors would like to thank the reviewer for the time spent in reviewing this manuscript and great feedback provided. I am sure this has helped this manuscript take a better shape. Please find our point-by-point response for your feedback.

I appreciate that this will probably be disheartening but it’s important for research to be reported as well as possible.

You might find it helpful to look at the Good Reporting of A Mixed Methods Study (GRAMMS) checklist. This is an old paper but it’s accessible and should indicate how you could improve the integration of data 

and consideration of the limitations (see, O'Cathain A, Murphy E, Nicholl J. The quality of mixed methods studies in health services research. J Health Serv Res Policy. 2008;13: 92-98).

The main issues include the following (further details are provided later in my comments).

  1. There are a lot of typographical and/or grammatical errors that need addressing

Response: Thanks for pointing this out. These are addressed now.

  1. The references are not contemporary, and although I appreciate some references are seminal and stand the test of time, there is only one reference as far as I can see that is 2020 or later,  7 date from 1999 or earlier and most are from the early 2010s.

Response: We have now included more contemporary references in our introduction and discussion sections.

  1. It’s not clear when the data were collected.

Response: Data was collected between 2013 and 2015. We have clarified that in the manuscript now.

  1. There is no evidence of data synthesis of the qualitative and quantitative data as would be expected in a mixed-methods study

Response: We have now integrated the quantitative data into the qualitative findings and how one explains the other.

  1. The qualitative findings have the potential to be interesting but the way they are presented, minimal text from the authors and lots of quotes is not what is expected of high-quality qualitative presentation. This section would benefit from editing and integrating the quotes more coherently.

Response: we have re-written the results section to now include more text and use quotes to support them.

  1. There are gaps in the paper that need addressing so that the reader can follow things through more easily.

Response: We have addressed the gaps identified by the reviewers to the best possible manner.

Further comments

  1. L94, when were the data collected?

Response: Data was collected between 2012 and 2015

  1. L 101 Need to add in a time point …. CSR interviews were completed at 101 baseline (0 months) and xxxxx (15 months)

Response: All the participants were not recruited at the same time hence this is not possible.

  1. L105 change are to were….. Health care professionals who were..

Response: Agreed and changed

  1. Table 1. Give numbers as well as % for Age, clinical designation,received advanced clinical certif. years of clinical training:

 Response: Agreed and changed

  1. L115 - approval number should be included.

Response: This information is included now in the manuscript

  1. L119 change sort to sorted.

Response: Agreed and changed

  1. l122, lower case for Guide.

Response: Agreed and changed

  1. l124, remove ‘about’.

Response: Agreed and changed

  1. l126… change start of sentence to maybe….. They were then asked a series of questions about how research would help managing their patients’ problem

Response: Agreed and changed

  1. l127 Note you need an apostrophe on patients’

Response: Agreed and changed

  1. L157 - pleases comments about reformatting table I have made in relation to Table 3.

Response: Agreed and changed. Table 3 is now table 2 and table 2 is now table 3. This is done to ensure that the tables are arranged in the order the results are presented.

  1. L76 and elsewhere. I’d suggest putting the participant label within brackets…  e.g.,  …….and then the comparison group” (PT 745).

Response: Agreed and changed

  1. You also need a full stop at the end of sentence/quote.

Response: Agreed and changed

  1. L182 and elsewhere - some labels are not linked to the type of participant, just a participant number. This needs to be addressed throughout.

Response: Agreed and changed. We have provided the labels now.

  1. Generally - it’s really hard to read the paper as the quotes just look like usual text. Something needs to be done (e.g., indenting quotes.

Response: The quotes are italicised now to help differentiate from the text.

  1. Also, I think there are many quotes where further condensing could happen. Typically you can weave short sections of quotes into text sentences rather than just describing something and then following it with a quote.

Response: Thanks for this suggestion. We done this to some of the quotes now.

  1. L167-8 delete ‘(population or patient, intervention or indicator, comparison or control, outcome)’ as you have previously provided this information.

Response: Agreed and removed.

  1. L180-181. Suggest you include numbers to support statements e.g. PubMed (n=x, %) was the most commonly used followed by Google Scholar (n=x, %). The other that were used are CINAHL (n=x, %), OVID (n=x, %), Psych Info (n=x, %).

Response: This information is included now

  1. L181 and elsewhere you write Psych info, but should Info be capitalised

Response: Agreed and changed

  1. L194 what is -651?

Response: This was a participant number that was not linked to the label. This information is included now.

  1. Figure 1 is fairly pointless - it’s hard to read and isn’t particularly helpful. I would delete this and consider a different way of presenting data. Maybe a table that is in supplementary data.

Response: The authors would like to keep this figure as it is a word cloud. In the image the greater the font size of the search engines included in the image the higher the frequency of them being used to search evidence to answer clinical questions. This information is now included in the manuscript.

  1. L200-201. The first sentence needs rephrasing.

Response: Agreed and rephrased now

  1. I’m not convinced you need a table for this. I think it could be easily summed up in text. BUT if you retain the table, I’d suggest reworking it so it becomes the same width as the text to make the table look professional in the final version.

Response: We have reworked the table so that it matches same width as the text.

  1. L252 and elsewhere - you make one point and then follow it with 3 quotes, this is not the best approach to qualitative reporting, you need to be selective in choice of ‘indented’ quotes and use the text to make points. Otherwise the reader is just reading a ’shopping list’ of quotes.

Response: we have now used more text to convey the points to the readers.

  1. L252 they not thy

Response: Agreed and changed

  1. L260 - is this a sub-sub heading?

Response: We have removed this as it did not make sense to have it there.

  1. L268 some quotes/sentences don’t make sense, e.g.. “Being that I have been a member of New Zealand Pain Society so frequently giving up dates theories different online”.

Response: This has been removed now.

  1. L282 Delete ‘A’

Response: This has been removed now.

  1. L292 revise to    The mean total CSR score w as 36 out of 52 (69.23%).

Response: Agreed and changed

  1. L292 - is there more you can present in terms of the quantitative results. The results are confusing and the reader needs to understand what you’re presenting…. You talk of the mean total score being 36 and then go on to talk of mean scores of 4-6 and 2-4… is this for individual items? I appreciate you have included a table earlier on but explain what you mean by low and well.

Response. Yes these scores are for individual items. We have moved the quantitative results to the top of the results section. We have also explained this clearly now.

  1. L295-6 IF you retain the content of this sentence, then revise punctuation and grammar etc to “……were explored, it was found that the participants scored better on basic evidence search skills (mean scores 4 to 6) but when it came to advanced skills such as appraising evidence or grading quality of evidence etc. they had lower scores (mean scores 2-4).

Response: We have incorporated the reviewer’s comments here and have retained the sentence.

  1. L295-6 ALSO - how come they could score 4 in both low and higher scores?

Response: we have now changed this to less than 4 and greater than 4.

  1. L296 here and in discussion, if you have used a true mixed methods approach you would have a section on data synthesis but it appears that the qualitative and quantitative data are just addressed separately.

Response: We have now integrated the quantitative data into the qualitative findings and how one explains the other.

  1. L299 This is retrospective recall of experiences………

Response: Agreed and changed

  1. L319 and elsewhere…. Capitalise S in Google Scholar

Response: Agreed and changed

  1. L353 BUT as I’ve pointed out earlier on you have participants scoring 4 in both higher and lower categories.

Response: we have now changed this to less than 4 and greater than 4.

  1. L372 - The claim that this is mixed methods (convergent parallel strategy - see l96) can’t be a strength as there is no evidence of data synthesis.

Response: We have now integrated the quantitative data into the qualitative findings and how one explains the other.

  1. L377- all references should be within the same set of brackets.

Response: Agreed and changed

  1. The Discussion is inadequate at the moment.  The main issue is that it is relying on old references and not making any comment about this. If the point is that findings in this study are similar to those from years ago, this is an important point. BUT there should be more contemporary literature to draw on and to contextualise. At the moment it feels somewhat confused and it’s difficult to determine if it could make a contribution.

Response: We have rewritten sections of the discussion with references that are recent.

Round 2

Reviewer 2 Report

I thank the authors for their feedback and I realize that they worked hard to make the study clearer and more robust.

The summary was changed and questions related to the results were inserted, but the conclusion is not very clear, in which the proposed objective must be answered in view of the results found.

In the introduction, the authors inserted some requested and pertinent questions. I notice that the suggestions were accepted and that the text is clearer and more concise.

The results were also changed, making them clearer and increasing the clarity of the findings. They also presented the necessary inferences and data integration.

The discussion was adjusted and limitations were presented, which increases the internal and external validation of the study. The conclusion is clear and objective. This needs to be addressed in the summary.

I wish the authors success and congratulate them on the study.

My greetings.

Author Response

The authors would like to thank the reviewer for their feedback.

I thank the authors for their feedback and I realize that they worked hard to make the study clearer and more robust. 

The summary was changed and questions related to the results were inserted, but the conclusion is not very clear, in which the proposed objective must be answered in view of the results found.

Response: We have now changed the conclusion to match the objective.

 In the introduction, the authors inserted some requested and pertinent questions. I notice that the suggestions were accepted and that the text is clearer and more concise.

 The results were also changed, making them clearer and increasing the clarity of the findings. They also presented the necessary inferences and data integration.

 The discussion was adjusted and limitations were presented, which increases the internal and external validation of the study. The conclusion is clear and objective. This needs to be addressed in the summary.

Response: We have now changed the conclusion to match the objective in the summary.

Reviewer 3 Report

L50 Maybe change ‘Literature search is essentially an art  and a skill that improves training and practice’ to ‘Literature searching is essentially an art  and a skill that improves with training and practice’ .

L66 The comparison between an electronic database and a print journal doesn’t really make sense. The database supports searching a whole range of literature (as well as providing access to articles), a print journal does not provide the search option (apart from within its own pages). I don’t think this sentence really adds anything and it could be deleted or you could just delete after ‘from home’.

l82, suggest you change ‘There are very minimal studies…’ to ‘There are few studies…’

L87, Knowledge and Physician do not need capitalisation.

L89 ‘and’ is missing …. in pain management and found that all of them….

L103-106 Rephrase this new sentence as it just doesn’t make sense……. Getting to know their experiences would enable them to plan better means of pushing out evidence to clinicians and modify current methods to provide a better experience of searching for evidence, thus improving evidence-based practice. 

L166, change ‘helped’ to ‘help’

L238-239 - seeing as the quote is so short it would be better if it was indented from the main text and just followed as part of the sentence.

The Figure of the word cloud adds nothing really to the paper. I would delete this - it’s a bit of a whimsical way of reporting in an academic paper. I would just simply report that the top 5 data bases were …., yyy, xxxx etc.

L502 I think cross section should read cross sectional.

L507. Maybe change literature search to literature searching

Author Response

The authors would like to thank the reviewer for their feedback

L50 Maybe change ‘Literature search is essentially an art  and a skill that improves training and practice’ to ‘Literature searching is essentially an art  and a skill that improves with training and practice’ .

Response: Agreed and changed

L66 The comparison between an electronic database and a print journal doesn’t really make sense. The database supports searching a whole range of literature (as well as providing access to articles), a print journal does not provide the search option (apart from within its own pages). I don’t think this sentence really adds anything and it could be deleted or you could just delete after ‘from home’.

Response: Agreed and changed

l82, suggest you change ‘There are very minimal studies…’ to ‘There are few studies…’

Response: Agreed and changed

L87, Knowledge and Physician do not need capitalisation.

Response: Agreed and changed

L89 ‘and’ is missing …. in pain management and found that all of them….

Response: Agreed and changed

L103-106 Rephrase this new sentence as it just doesn’t make sense……. Getting to know their experiences would enable them to plan better means of pushing out evidence to clinicians and modify current methods to provide a better experience of searching for evidence, thus improving evidence-based practice. 

Response: We have clarified this sentence now.

L166, change ‘helped’ to ‘help’

Response: Agreed and changed

L238-239 - seeing as the quote is so short it would be better if it was indented from the main text and just followed as part of the sentence.

Response: Agreed and changed

The Figure of the word cloud adds nothing really to the paper. I would delete this - it’s a bit of a whimsical way of reporting in an academic paper. I would just simply report that the top 5 data bases were …., yyy, xxxx etc.

Response: Agreed and changed. We have removed the word cloud and have reported the top databases.

L502 I think cross section should read cross sectional.

Response: Agreed and changed

L507. Maybe change literature search to literature searching

Response: Agreed and changed